# Bacteriophage-Derived Double-Stranded RNA Exerts Anti-SARS-CoV-2 Activity In Vitro and in Golden Syrian Hamsters In Vivo

**DOI:** 10.3390/ph15091053

**Published:** 2022-08-25

**Authors:** Kristine Vaivode, Irina Verhovcova, Dace Skrastina, Ramona Petrovska, Madara Kreismane, Daira Lapse, Zane Kalnina, Kristine Salmina, Diana Rubene, Dace Pjanova

**Affiliations:** Latvian Biomedical Research and Study Centre, Ratsrtupites Street 1, k-1, LV-1067 Riga, Latvia

**Keywords:** COVID-19, SARS-CoV-2, antiviral drug, dsRNA, bacteriophage-derived dsRNA

## Abstract

Bacteriophage-derived dsRNA, known as Larifan, is a nationally well-known broad-spectrum antiviral medication. This study aimed to ascertain the antiviral activity of Larifan against the novel SARS-CoV-2 virus. Larifan’s effect against SARS-CoV-2 in vitro was measured in human lung adenocarcinoma (Calu3) and primary human small airway epithelial cells (HSAEC), and in vivo in the SARS-CoV-2 infection model in golden Syrian hamsters. Larifan inhibited SARS-CoV-2 replication both in vitro and in vivo. Viral RNA copy numbers and titer of infectious virus in the supernatant of Calu3 cells dropped significantly: *p* = 0.0296 and *p* = 0.0286, respectively. A reduction in viral RNA copy number was also observed in HSAEC, especially when Larifan was added before infection (*p* = 0.0218). Larifan markedly reduced virus numbers in infected hamsters’ lungs post-infection, with a more pronounced effect after intranasal administration (*p* = 0.0032). The administration of Larifan also reduced the amount of infections virus titer in the lungs (*p* = 0.0039). Improvements in the infection-induced pathological lesion severity in the lungs of animals treated with Larifan were also demonstrated. The inhibition of SARS-CoV-2 replication in vitro and the reduction in the viral load in the lungs of infected hamsters treated with Larifan alongside the improved lung histopathology suggests a potential use of Larifan in also controlling the COVID-19 disease in humans.

## 1. Introduction

In December 2019, a series of pneumonia cases of unknown cause emerged in Wuhan, China, with clinical presentations greatly resembling viral pneumonia. This disease was caused by a novel form of coronavirus named ‘severe acute respiratory syndrome coronavirus 2′ (SARS-CoV-2), and the disease was accordingly named the coronavirus disease 2019 (COVID-19). On 11 March 2020, the World Health Organization announced this emerging disease as a fast-progressing pandemic. COVID-19 is a complex disorder, and its manifestations can range from mild upper respiratory illness to severe bilateral pneumonia, acute respiratory distress syndrome, disseminated thrombosis, multi-organ failure, and death [1,2].

Since the emergence of this disease, numerous anti-COVID-19 vaccines have been tested in clinical trials worldwide. Several have been approved in record time and are currently being successfully used to immunize people. However, vaccination effectiveness is hindered by the emergence of novel virus variants [3,4]. Monoclonal antibody therapies are costly, and the best anti-COVID-19 drug approved so far—Remdesivir—showed only a mild effect against the SARS-CoV-2, slightly reducing the hospitalization time of patients [5]. Another recently approved antiviral drug against COVID-19, Paxlovid, also loses its effectiveness in the case of novel virus mutants already circulating in infected people [6]. This raises the concern that physicians could soon lose one of the best available therapies. Thus, there is still an urgent need for antiviral drugs that could slow down the replication of SARS-CoV-2 and fulfill an essential role in treating COVID-19 patients, e.g., preventing or alleviating disease symptoms. In addition, antiviral drugs could be used as a preventive measure to protect high-risk groups, especially if they could be administered in a non-invasive manner.

Here, we present a nationally well-known antiviral drug in the form of double-stranded RNA (dsRNA) as a potential anti-COVID-19 medication. Bacteriophage-derived dsRNA, also known as Larifan (Larifan Ltd., Riga, Latvia), is a heterogeneous population of dsRNAs. It has been obtained biotechnologically from *E. coli* cells infected with f2sus11 amber mutant bacteriophage and comprises dsRNA molecules (*acidum ribonucleinicum duplicatum*) with an average length of 700 base pairs. Larifan has been developed as a poly-functional and wide-spectrum antiviral drug that is a potent inducer of endogenous type I interferons (IFNs). At the time of its invention, the mechanism of Larifan’s action was studied by Sokolova et al. in both experimental systems [7] and clinical trials on volunteers [8]. The lateral results proved Larifan’s ability to induce and activate enzymes of the IFN system, particularly dsRNA-activated protein kinase R (PKR) and 2′-5′ oligoadenylate synthase (OAS), both involved in the translation blockage in virus-infected cells [9]. Currently, Larifan is approved and registered for human use at the State Agency of Medicines of the Republic of Latvia as a treatment method for herpes virus infections and secondary immunodeficiency (Reg. No.04-0230).

COVID-19 is associated with a delayed type I IFN response [10]. In the case of other coronavirus family members, e.g., SARS and MERS, the administration of exogenous type I IFNs reduced disease symptoms [11], which is considered a promising therapeutic option for COVID-19 as well [12,13]. Moreover, SARS-CoV-2 has developed mechanisms for evading its recognition by the cellular pathogen recognition receptors (PRRs) [14,15,16]. By applying a PRR agonist such as Poly (I:C), a well-known synthetic dsRNA, the innate immune inhibitory effect of the virus can be overruled, especially at an early stage during infection [17].

In the present study, we evaluated the potential to inhibit SARS-CoV-2 replication in the presence of bacteriophage-derived dsRNA in vitro in human lung adenocarcinoma cell line (Calu3), primary human small airway epithelial cells (HSAEC), and in vivo in an infection model of golden Syrian hamsters. It has been shown that golden hamsters are suitable for COVID-19 research, as features associated with the SARS-CoV-2 infection in golden hamsters resemble those found in humans with the presence of high virus titer until day five after infection. Nevertheless, all animals fully recover within 14 days of infection [18]. Promisingly, Larifan exerts an antiviral effect against SARS-CoV-2 in vitro and in vivo.

## 2. Results

### 2.1. Replication of SARS-CoV-2 in Calu3 and HSAEC Cells

The cytolysis of cells was evident only in Calu3 cells (Figure 1A,B). Viral RNA copy numbers in the supernatant of Calu3 cells were abundant and peaked on days three and four post-infection. After that peak, the viral RNA copy number decreased (Figure 1C). In HSAEC, SARS-CoV-2 infection did not elicit the typical cytopathic effect (CPE) (Figure 1D,E) that is characteristic for many respiratory viruses; however, cells appeared morphologically to “inflate”, and virus release within supernatant could be measured with the increase in viral RNA copy number even until day six post-infection (Figure 1F). This suggests a different and prolonged replication mode of SARS-CoV-2 in HSAEC that does not produce apparent damage to the cells but allows continuous virus release. Of note, the infection of HSAEC by an enterovirus exhibited the typical CPE (Figure 1G).

### 2.2. Larifan Exserts an Antiviral Activity against SARS-CoV-2 In Vitro

Larifan showed unapparent cytotoxicity in Calu3 and HSAEC cell lines at all the concentrations used in this study. The half-maximal cytotoxic concentrations of Larifan in Calu3 cells and HSAEC were 985.95 µg/mL and 3962.18 µg/mL, respectively (Figure 2A). As expected, SARS-CoV-2 infection caused a very high increase in virus RNA copy numbers and infectious virus titer in both cell lines under investigation. This is in line with others’ observations, who also noticed an abundant presence of the virus in common cell culture models [19,20]. Despite the high virus presence, Larifan in both cell lines inhibited the replication of SARS-CoV-2. Viral RNA copy numbers in the supernatant of Calu3 cells dropped from 9.6 × 10^6^ to an average of 1.4 × 10^6^ (*p* = 0.0296) when Larifan was present. Out of the three conditions analyzed, the virus RNA copy number dropped significantly if Larifan was added pre- (*p* = 0.018) and post-infection (*p* = 0.0054). In contrast, no significant drop could be observed in the case of full-time treatment (*p* = 0.6204) (Figure 2B). The titer of the infectious 7.0log_10_ virus measured in Vero E6 cells also dropped significantly in Calu3 cell supernatant to an average of 4.5log_10_ (*p* = 0.0286) when Larifan was present. Furthermore, there was a significant decrease in the virus titer when Larifan was added before infection (*p* = 0.0014) (Figure 2C). A reduction in viral RNA copy number was also observed in HSAEC (Figure 2D), especially when Larifan was added before infection (*p* = 0.0218) with a drop from 4.1 × 10^5^ to 2.3 × 10^4^ accompanied by a decrease in virus presence in the cells assessed by immunofluorescence of SARS-CoV-2 nucleoprotein (NP) (Figure 2E–G).

### 2.3. Administration of Larifan Decreases SARS-CoV-2 Virus Numbers In Vivo

The experimental strategy of the study is depicted in Figure 3A. SARS-CoV-2-infected hamsters treated with Larifan either s.c. or i.n. presented a decrease in both viral RNA copies and infectious virus titer per mg of lung tissue compared with untreated hamsters. Such an effect was observed on days three and five post-infection, with more pronounced differences on day three. Intranasal Larifan administration gives a reduction in 2.7log_10_ and 2.0log_10_ RNA copies per mg of lung tissue at days three and five post-infection, respectively, which significantly differs from the viral numbers measured without the presence of Larifan (*p* = 0.0032) (Figure 3B). Infectious virus titers in the lung also dropped by 4.3log_10_ TCID_50_ on day three, and by 2.8log_10_ TCID_50_ on day five post-infection, which also significantly differed from the untreated group (*p* = 0.0039) (Figure 3C). Similarly, Larifan administration s.c. also resulted in the detection of fewer virus RNA copies and smaller virus titers in the lung. Although, these changes were less pronounced with the drop of 1.7log_10_ RNA copies per mg of lung tissue on day three post-infection, and no differences were observed on day five (Figure 3B). The decline of infectious virus titers was also less pronounced in the case of s.c. administration and was 2.3log_10_ TCID_50_ on day three and 1.5log_10_ TCID_50_ on day five (Figure 3C). A more pronounced effect was measured when Larifan was administrated i.n. reaching statistical significance between administration modes, particularly in the number of RNA copies measured (*p* = 0.027) (Figure 3B).

Intranasal administration of Larifan caused slightly higher weight loss in treated animals during the first days after infection than in untreated animals. However, on day five, the opposite effect was observed, and Larifan-treated hamsters had lost less weight when compared to virus-infected and untreated animals (Figure 3D). However, these changes were not statistically significant and might relate to Larifan’s systemic effect on the living organism as it mimics viral infection. Weight gain on day five might indicate faster recovery. Subcutaneous administration of Larifan caused minor weight loss and was comparable with that of untreated hamsters or even less. On day five, the weight loss of treated animals was noticeably smaller than that of untreated hamsters (Figure 3D).

### 2.4. Administration of Larifan Reduces SARS-CoV-2 Antigen in Lungs and Improves Histological Lung Pathology

The lung immunocytochemistry for SARS-CoV-2 NP showed that the viral antigen on day three post-infection was predominantly located in the bronchiolar epithelium (Figure 4A). On day five post-infection, immunoreactivity was largely cleaned from the bronchiolar epithelium and was observed in different alveolar regions displaying a patchy distribution pattern (Figure 4B). In Larifan-treated hamsters, a marked reduction in SARS-CoV-2 NP positive cells was observed (Figure 4C–F). The decrease in SARS-CoV-2 NP was evident both on days three and five post-infection and after s.c. and i.n. Larifan administration; however, the scored assessment of NP expression reached statistical significance only in the case of i.n. Larifan administration, *p* = 0.0026 and *p* = 0.0161 on days three and five post-infection (Figure 4M).

Alongside the reduction in SARS-CoV-2 NP expression, improvements in the infection-induced pathological lesion severity were observed in hamster lungs. Infected hamster lungs show several pathological changes that include thickening of the interalveolar septa predominantly due to inflammatory infiltrate, hyperemia, and vascular thrombosis. Alveolar septum fibrosis and signs of bronchial epithelium hyperplasia were also observed (Figure 4G,H). Noteworthy, in the lungs of Larifan-treated hamsters, either s.c. or i.n., these changes were less pronounced and predominantly focal (Figure 4I–L) and statistically significant only in the case of i.n. Larifan administration, *p* = 0.0009 and *p* = 0.0144 on days three and five post-infection (Figure 4N).

## 3. Discussion

COVID-19, an acute contagious respiratory disease caused by SARS-CoV-2 with high infectivity, is a worldwide concern. The clinical course of the disease showed that patients experienced a latent infection for one to two weeks, with a high possibility of transmitting the infection to other individuals [21]. This suggests a possible distinct infection process of this virus. Most respiratory viruses, like influenza and rhinovirus, in vitro elicit a typical CPE in virus-infected cells [22,23], whereas SARS-CoV-2 does not [24]. This study by Liao et al. noticed a continuous release of virus particles from the human bronchial epithelial cells without cell damage [24]. We have made similar observations in HSAEC used as a model epithelial cell line in our study. We have also observed high numbers of virus RNA copies in cell cultures under investigation, which is in line with previous observations [19,20]. Collectively, these data show that SARS-CoV-2 uses a “clever” infection strategy that might be linked to the asymptomatic disease and high prevalence. This means that suitable preventive countermeasures should be considered. Therefore, antiviral drugs slowing down the replication of SARS-CoV-2 would be clinically beneficial.

Larifan, as an antiviral drug at the time of its invention, was shown to be effective against many different viruses. Larifan inhibited the reproduction of cytomegalovirus [25] and the replication of human immunodeficiency virus-1 [26] in cell cultures. The drug worked in immunocompromised mice infected with herpes simplex virus (HSV)-1 and HSV-2 [27] and prolonged the life span of monkeys infected with smallpox [28]. In addition, the drug demonstrated high antiviral efficacy against Omsk hemorrhagic fever in laboratory animals [29] and was effective against influenza [30] and rabies [31]. In a clinical setting, Larifan appeared effective against acute herpetic stomatitis in children [32], genital papillomavirus infections in women [33], and combined with the herpetic vaccine, led to an improvement of the clinical symptoms of recurrence [34]. Moreover, since its launch for use in human patients, it has been successfully used in the clinic predominantly against HSV infections with a safety track record.

Here, we demonstrate that the presence of Larifan reduces the replication of SARS-CoV-2 both in vitro, in Calu3 cells and HSAEC, and in vivo in the SARS-CoV-2 infection model of golden Syrian hamsters. Although we have observed an inhibitory effect of Larifan on SARS-CoV-2 replication, the underlying mechanism of its antiviral effects with modern state-of-the-art methods is still to be determined. We propose that Larifan mimics viral effects targeting the host and preventing further virus replication, including invading viruses, and, thus, might have a dual action. First, local and direct antiviral activity at the cellular level is realized through the activation of enzymes of the IFN system, PKR, and OAS, which leads to the global translation blockage of both cellular and viral mRNA in virus-infected cells [7,8], and second, through interfering with the host immune reaction [35,36], leading to the early activation of the innate immune system and elimination of invading pathogens.

Indeed, we have observed Larifan’s inhibitory effect on SARS-CoV-2 at the cellular level in vitro and in vivo. Surprisingly, full-time Larifan treatment in vitro was less effective than solely pre-treatment with Larifan or treatment immediately after infection. We hypothesized that this might be related to Larifan’s ability to activate PKR. A defining feature of PKR is the “bell-shaped” curve for activation where low concentrations of dsRNA activate but higher concentrations are inhibitory. As Lamaire et al. postulated, “*These results can be rationalized in a model where low concentrations of dsRNA favor assembly of multiple proteins—possibly assembling as dimers—on a single dsRNA whereas higher dsRNA concentrations dilute PKR monomers onto separate molecules of dsRNA*” [37]. It seems that at the cellular level, the addition of Larifan repeatedly causes the inhibition of PKR and a subsequent decrease in Larifan’s inhibitory effect on SARS-CoV-2. Of note, in the case of full-time treatment, a “bell-shaped” response is evident. Such a dose dependency is not observed in the case of single pre-treatment or single treatment only after infection. This might be explained by the fact that for this study, we have chosen concentrations of Larifan that fit the activation window’s “bell dome” and that were selected based on dynamics of type I IFN production in peripheral blood mononuclear cells after Larifan treatment [24]. Since treatments that interact with the host are postulated to be effective in the early stages of the disease, we administered Larifan before the infection to evaluate its prophylactic potential. Additionally, Larifan treatment immediately after infection as a so-called emergency prophylactic usage was done to model a situation where an infection has occurred, but no signs of disease are yet evident. Nonetheless, the effects of post-infection treatment later during the disease course need to be clarified to prove the hypothesis that such a treatment has no benefit.

For in vivo experiments, we, however, stick to a treatment regime with Larifan administration repeatedly for two reasons: (i) to be sure that the immune system does not eliminate Larifan after a single dosage and its effect can be observed, and (ii) treatment regime in humans also involves repeated administration of Larifan (four consecutive injections with an interval of 24 h).

As it seems evident that Larifan acts both at the cellular level and interferes with the immune system, we have also chosen two different ways of Larifan administration in vivo, namely i.n., that represents a local and more direct mode of action, and s.c., corresponding to systemic effect. In both situations, Larifan inhibited the virus infection. More considerable inhibition was observed on day three. On day five, the number of virus copies again increased; however, it was still significantly lower than in the untreated hamsters. Whether this means that the drug has a short time effect remains to be answered by following hamsters for a longer time because the presence of high virus titer and signs of the disease in hamsters persists until day five post-infection. After this point, all animals fully recover within 14 days [18]. None of the Larifan-treated hamsters showed signs of fever on day five in comparison with untreated hamsters (data not shown). It is worth pointing out that i.n. admission of Larifan inhibited the virus infection better than the systemic administration in the s.c. experiment model. These results might suggest that Larifan acts better locally in case of SARS-CoV-2 infection. A more local action means that Larifan could be potentially administered in a non-invasive manner, e.g., as a nasal spray. In this form of administration, the medication could be utilized for preventative measures in high-risk people and also as a treatment to reduce the virus numbers in infected individuals, especially in the early stages of the disease, as it would promote faster recovery.

Aforesaid, the type I IFN response in a SARS-CoV-2 infection is hampered [10], resulting in rapid virus replication in pneumocytes, cytokine, chemokine induction, and inflammatory monocyte/macrophage recruitment to the lungs [14,15]. The delayed innate-immune response provides a “time window” for viral replication and, eventually, cytokine storm and death [11].

Exogenous type I IFN therapy has been used previously in clinical trials as a treatment for several coronavirus types [12,13], therefore, it could also prove to be beneficial in treating COVID-19. One of the reasons for this is that SARS-CoV-2 has evolved mechanisms to avoid cellular PRR recognition, in particular, melanoma differentiation-associated protein-5 (MDA5) that belongs to the innate immune system and participates in the detection of viral invasions [16,38,39]. Thus, the application of PRR agonists like MDA5 and toll-like receptor 3 agonist Poly (I:C) could potentially counteract the negative effect that the virus exerts on the immune system [17].

It should be mentioned that clinical trials are ongoing to examine the therapeutic potential of Poly (I:C) against SARS-CoV-2 (https://clinicaltrials.gov/ct2/show/NCT04672291) (accessed on 8 July 2022). Larifan, just like Poly (I:C), is a dsRNA of bacteriophage origin with a similar action [36], and, thus, might be promising in treating COVID-19. Although we have demonstrated here an antiviral effect of Larifan against SARS-CoV-2 in vitro and in hamster SARS-CoV-2 model in vivo, the innate immune response to Larifan treatment first in model systems, as well as the best timing for such a therapy, still remains to be elucidated.

## 4. Materials and Methods

### 4.1. SARS-CoV-2

The SARS-CoV-2 strain used in this study, SARS-CoV-2 hCoV-19/Sweden/20-53846/2020 (lineage B 1.1.7, UK), was obtained from the European Virus Archive and propagated in Vero E6 cells; a passage four virus was used for the studies described here. The titer of the virus stock was determined by a 50% tissue culture infective dose (TCID_50_) according to the cytopathic effect using the Reed–Muench method. All the infection experiments were performed in a biosafety level-3 (BSL3) laboratory.

### 4.2. Cell Lines

Vero E6 cells (African green monkey kidney, ATCC CRL-1586) and Calu3 (human lung adenocarcinoma cells, ATCC HTB-55) were cultured in Dulbecco’s Modified Eagle Medium (DMEM) supplemented with 10% fetal bovine serum (FBS), 1% l-glutamine, and 1% bicarbonate (all from Gibco, Carlsbad, CA, USA). HSAEC (primary human small airway epithelial cells, ATCC PCS-301-010) were grown in an Airway Epithelial Cell Basal Medium (ATCC PCS-300-030) supplemented with a Bronchial Epithelial Cell Growth Kit (ATCC PCS-300-040). Virus end-point titrations in Vero E6 cells were performed with a medium containing 2% FBS instead of 10%. All cells were incubated in a humidified atmosphere at 37 °C with 5% CO_2_.

### 4.3. Compound

Larifan was purchased from Larifan Ltd. According to the manufacturer, it is obtained from E. coli cells infected with f2sus11 amber mutant bacteriophage and represents a heterogeneous population of dsRNAs with an average length of 700 base pairs in a physiological solution. For experiments, 10 mg/mL stock solution was diluted with appropriate cell culture media.

### 4.4. Cytotoxicity Assay

The cytotoxic effects of Larifan on Calu3 and HSAEC cells were evaluated by the Cell Counting Kit-8 (CCK8—Dojindo Laboratories). Monolayers of Calu3 and HSAEC cells in 96-well plates were incubated with indicated concentrations of Larifan. After 72 h, CCK8 solution was added, and cells were incubated for an additional 4 h at 37 °C with 5% CO_2_. The absorbance was measured at 450 nm. Half-maximal cytotoxic concentration (CC50) was calculated according to the best fit point-to-point line’s interpolated value.

### 4.5. Infection of Cells with SARS-CoV-2 and Treatment with Larifan

Calu3 and HSAEC cells were grown to a monolayer in 6-well plates. Cells were infected with SARS-CoV-2 at a multiplicity of infection (MOI) of 0.3. Samples for virus assessment were collected every 24 h. Cells were treated with varying final concentrations (50, 100, 200 and 400 µg/mL) of Larifan in the medium. Cells were pre-treated with Larifan for 4 h, and the virus was then added to allow attachment for 1 h. Afterwards, the culture medium was removed, and a fresh drug-containing medium was added, in which cells were cultured until the end of the experiment (“full-time treatment”). For pre-infection treatment, Larifan was added to the cells for 4  h before viral attachment. Then virus containing media was removed, and cells were cultured in a fresh medium without the drug until the end of the experiment. For post-infection treatment, Larifan was added to the fresh medium immediately after virus infection, and cells were cultured in this medium with the drug until the end of the experiment. Following 72 h of incubation (day three p.i.), samples were collected for virus assessment.

### 4.6. SARS-CoV-2 Infection Model in Hamsters

We used the hamster infection model of SARS-CoV-2 as described previously [40]. The experimental procedures in animals were approved by the National Animal Welfare and Ethics Committee (permit no. 124/2021) and were performed in compliance with Directive 2010/63/EU as adopted in the national legislation.

In total, 24 naïve 9–10-week-old specific pathogen-free (SPF) male golden Syrian hamsters (strain HsdHan^®^: Aura) were purchased from Envigo (Indianapolis, IN, USA) and, after introduction, were single-housed in individually ventilated cages GR900 (Tecniplast, Buguggiate, Italy), HEPA-ventilated by SmartFlow air handling unit (Tecniplast, Buguggiate, Italy) at 75 air changes per hour in a negative pressure mode. Access to autoclaved water acidified to pH 2.5–3.0 with hydrogen chloride and standard rodent diet (4RF21 (A), Mucedola, Settimo Milanese, Italy) was provided ad libitum. Aspen wooden bedding and nesting material (Tapvei, Harjumaa, Estonia) together with rat cardboard houses (Velaz, Praha, Czech Republic) and aspen gnawing bricks (Tapvei, Harjumaa, Estonia) were provided in all cages. Animals were housed in SPF facility BSL3 unit under controlled temperature (24 ± 1 °C) and relative humidity of 40–60%. All animals were subjected to at least a 7-day acclimatization period with a 7:00 am.–7:00 pm. visible light cycle. An individual animal served as an experimental unit in these experiments.

For the COVID-19 modeling, the isoflurane-anesthetized animals were intranasally infected with SARS-CoV-2. Briefly, animals were induced with 3–4% isoflurane and maintained under 2% isoflurane while inoculated intranasally with 100 µL containing 2 × 10^4^ TCID_50_ of the virus (50 µL per nostril).

Animals received the drug treatment either subcutaneously (s.c.) or intranasally (i.n.) using Larifan dosage of 5 mg/kg (taken from previous studies in animal models [14,15,16,17,18,19,20]) in phosphate-buffered saline. Hamsters were pre-treated with Larifan twice before virus infection (the second drug administration four hours before infection) and twice after virus infection with an interval of 24 h between each drug administration. Each experimental group (SARS-CoV-2, SARS-CoV-2 plus Larifan i.n. and SARS-CoV-2 plus Larifan s.c.) consisted of eight hamsters.

The group size was estimated using the resource equation approach [41]. Hamsters were monitored daily for clinical signs, appearance, behavior, and weight. On days three and five p.i., four hamsters from each group were humanely euthanized using 5% isoflurane anesthesia. Lungs were collected for histological observations and virus quantification by ddPCR and end-point titration.

### 4.7. RNA Isolation and Viral RNA Quantification by Digital Droplet PCR (ddPCR) Analysis

Viral RNA from cell culture supernatants was isolated as per manufacturer’s instructions using QIAamp MinElute Virus kit (Qiagen, Hilden, Germany). Cells were suspended in TRI Reagent (Sigma Aldrich, Darmstadt, Germany), and total RNA was extracted according to the manufacturer’s protocol. Hamster lungs were collected and homogenized using bead disruption with Lysing Matrix E from MP Biomedicals (US) either in TRI Reagent or DMEM for virus copy number and infectious virus detection, respectively, and centrifuged to pellet cell debris. Viral copy numbers were assessed via digital droplet PCR (ddPCR) using the SARS-CoV-2 ddPCR kit (#12013743) from Bio-Rad designed for specific detection of the 2019-nCoV with N1 and N2 primers (2019-nCoV_N1 Forward Primer: GAC CCC AAA ATC AGC GAA AT; 2019-nCoV_N1-Reverse Primer: TCT GGT TAC TGC CAG TTG AAT CTG; 2019-nCoV_N2-Forward Primer: TTA CAA ACA TTG GCC GCA AA; 2019-nCoV_N2-Reverse Primer GCG CGA CAT TCC GAA GAA) and probes (2019-nCoV_N1-P: FAM-ACC CCG CAT TAC GTT TGG TGG ACC-BHQ1; 2019-nCoV_N1-P: FAM-ACC CCG CAT/ZEN/TAC GTT TGG TGG ACC-3IABkFQ; 2019-nCoV_N2-P: FAM-ACA ATT TGC CCC CAG CGC TTC AG-BHQ1; 2019-nCoV_N2-P: FAM-ACA ATT TGC/ZEN/CCC CAG CGC TTC AG-3IABkF) targeting the nucleocapsid genes. Droplets were generated using Bio-Rad QX200 Droplet Generator and analyzed with QX200 Droplet Reader (Bio-Rad). Absolute quantifications of virus RNA copies were estimated by modeling a Poisson distribution using QuantaSoftTM analysis software version 1.7 (Bio-Rad). Mean signal from N1 and N2 was extrapolated and expressed as viral RNA copies per microliter of supernatant, number of cells, or milligrams of tissue.

### 4.8. End-Point Virus Titration

Cell culture supernatants and lung homogenates were put directly onto Vero E6 cells. End-point titrations were performed on confluent Vero E6 cells in 96-well plates. Viral titers were calculated by the Reed–Muench method and expressed as TCID50 per microliter of supernatant, number of cells, or milligrams of tissue.

### 4.9. Immunofluorescence Detection of SARS-CoV-2 in HSEAC and Hamster Lungs

For SARS-CoV-2 NP immunocytochemical staining, HSEAC were grown on glass coverslips. The cells were fixed in 10% neutral buffered formalin for 1 h and washed three times with phosphate-buffered saline (PBS). Then the cells were permeabilized with 0.5% Triton X100 in PBS and blocked for 15 min in PBS, 20% Tween, and 1% bovine serum albumin (BSA) at room temperature. Cells were then stained with SARS-CoV-2 NP primary antibody (Thermo Fisher, Waltham, US) overnight at 4 °C, then washed and stained with Alexa Fluor 594 conjugated secondary antibody (Abcam, Cambridge, UK). Cells were then embedded in ProLong Gold Antifade Mountant with DAPI (Thermo Fisher) and evaluated using Leica DM6000 fluorescence microscope (Germany).

### 4.10. Histology

Hamster lungs were fixed in 4% formaldehyde immediately post-mortem and embedded in paraffin. Tissue sections (7 µm) were stained with hematoxylin and eosin stains and analyzed blindly for lung damage by at least two independent pathologists. Similar to others [34,42], a specific modified scoring system was used with maximum of 8 points for lung damage evaluation: samples containing focal and/or diffuse inflammation were given a score from 0 to 3, where 0 represents no inflammation; 1—mild; 2—moderate; 3—severe. Samples containing both focal and diffuse inflammation were given an extra 1 point. In addition, perivascular and peribranchial inflammation were evaluated, and, if both were present, an additional 1 point was given.

### 4.11. Immunofluorescence Detection of SARS-CoV-2 in Hamster Lungs

Formalin-fixed paraffin-embedded tissues were cut into 3–4 μm thick sections using Leica microtome and mounted on Superfrost Plus slides (Thermo Fisher Scientific, J1800AMNZ). Sections were deparaffinized using xylene, then rehydrated in decreasing ethanol concentrations (96–50%). Tissue sections were permeabilized and antigen retrieval was performed using sodium citrate buffer with repeated heating. Tissue sections were then blocked with 2% BSA and stained with primary antibody against SARS-CoV-2 Nucleocapsid (Thermo Fisher) at 4 °C overnight. Next, sections were stained with Alexa Fluor 488 conjugated secondary antibody (Thermo Fisher). To reduce autofluorescence, CuSO4/NH4Cl solution was used. Cells were embedded in ProLong Gold Antifade Mountant with DAPI (Thermo Fisher) and evaluated using DM6000 fluorescence and laser scanning confocal SP8 microscopes (both Leica, Wetzlar, Germany). The intensity of immunostaining reaction for SARS-CoV-2 NP was analyzed using a 4-point semiquantitative grading scale adapted from [34] as follows: 0—no positive immunoreaction; 1—very mild; 2—mild; 3—moderate; 4—severe.

### 4.12. Statistical Analysis

GraphPad Prism (GraphPad Software, Inc., San Diego, US) was used to perform statistical analysis. Statistical significance in vitro between SARS-CoV-2 and Larifan groups was done using the Mann–Whitney ranks test, and Kruskal–Wallis tests with Dunn’s multiple comparison test were used for individual analysis between the groups. Two-way ANOVA was used when comparing the in vivo groups. For lung damage and immunohistochemistry scoring analysis, again Kruskal–Wallis tests with Dunn’s multiple comparison test were used. P values of ≤0.05 were considered significant.

## 5. Conclusions

In conclusion, the inhibition of SARS-CoV-2 replication in vitro and the potential reduction in the viral load in the lungs of hamsters treated with Larifan, alongside improved lung histopathology, suggests a potential benefit of this drug in humans. However, further studies are needed to elucidate the mechanism of Larifan’s action and continue with clinical studies to confirm its effectiveness and safety in humans suffering from the novel coronavirus disease. Noteworthy, new pathogenic viruses are expected to emerge in the coming decades, and proper knowledge and tools, including effective antivirals, will be in high demand.

## Figures and Tables

**Figure 1 pharmaceuticals-15-01053-f001:**
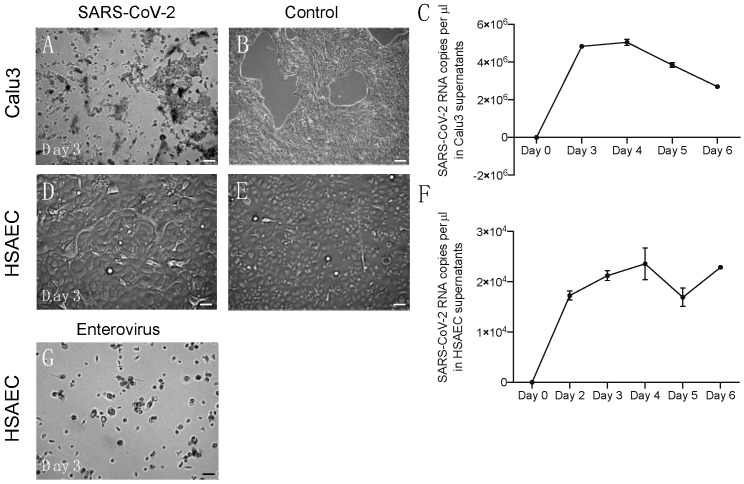
Virus replication in vitro in Calu 3 cells and HSAEC. (**A**) CPE observed in infected Calu3 cells on day three post-infection. (**B**) Morphology of uninfected Calu3 cells. (**C**) Viral RNA copy number measurements performed in duplicate in Calu3 cell supernatants. (**D**) Morphology of HSAEC on day three post-infection; no CPE observed. (**E**) Morphology of uninfected HSEAC. (**F**) Viral RNA copy number measurements performed in duplicate in HSAEC supernatants. (**G**) HSAEC infected with enterovirus; CPE observed at day three post-infection. Scale bars 100 µm.

**Figure 2 pharmaceuticals-15-01053-f002:**
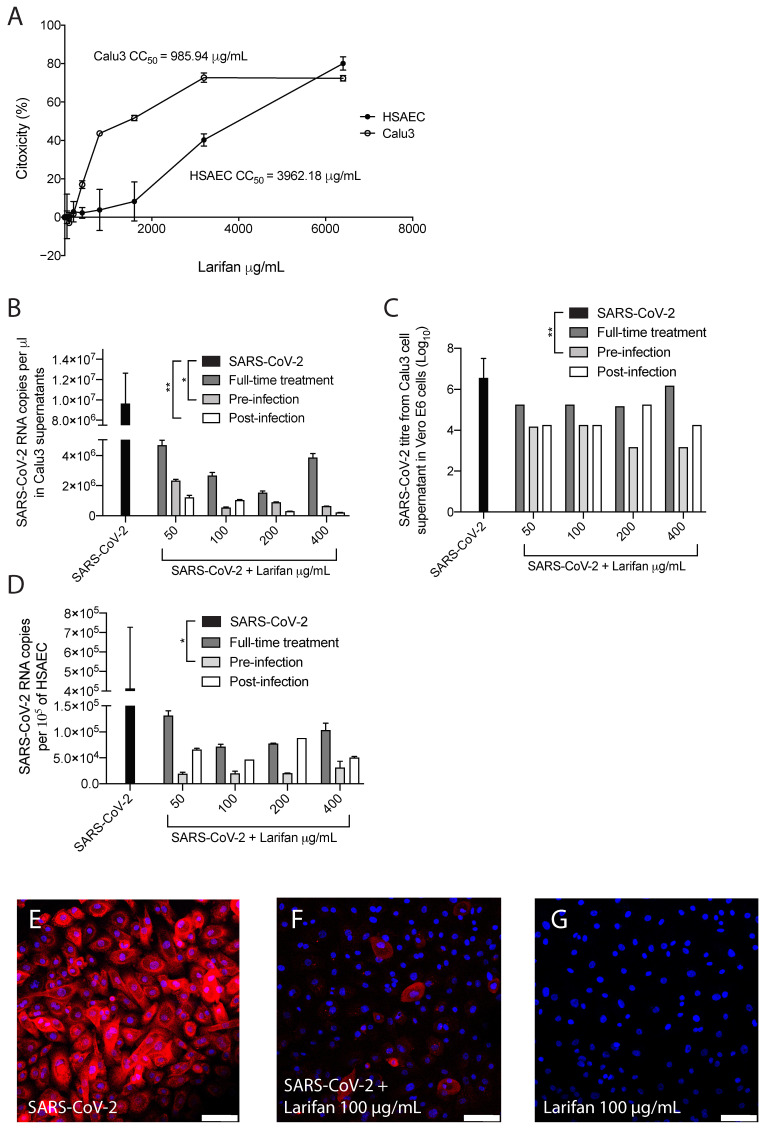
The antiviral activity of Larifan against SARS-CoV-2 in vitro. (**A**) No significant cytotoxicity of Larifan observed in Calu3 cells or HSAEC done in triplicate. (**B**) Viral RNA copy number drop in Calu3 cell supernatants at day three post-infection when Larifan was present throughout the experiment (“full-time treatment”) or added pre- or post-infection and performed in duplicate. (**C**) Infectious SARS-CoV-2 titer drop in Calu3 cell supernatants in paired experiments in Vero E6 cells after treatment with Larifan at day three post-infection when Larifan was present throughout the experiment or added pre- or post-infection. (**D**) Viral RNA copy number drop in HSAEC when Larifan was present throughout and added pre- or post-infection performed in duplicate. Data presented as mean (±SD). Statistical significance analysis was performed by using Kruskal–Wallis tests with Dunn’s multiple comparison test. * *p* < 0.05, ** *p* < 0.01. (**E**–**G**) Representative pictures of SARS-CoV-2 NP (red) detection by immunocytochemistry in HSAEC upon treatment with 100 µg/mL Larifan. Nuclei are counterstained with DAPI. Scale bars 100 µm.

**Figure 3 pharmaceuticals-15-01053-f003:**
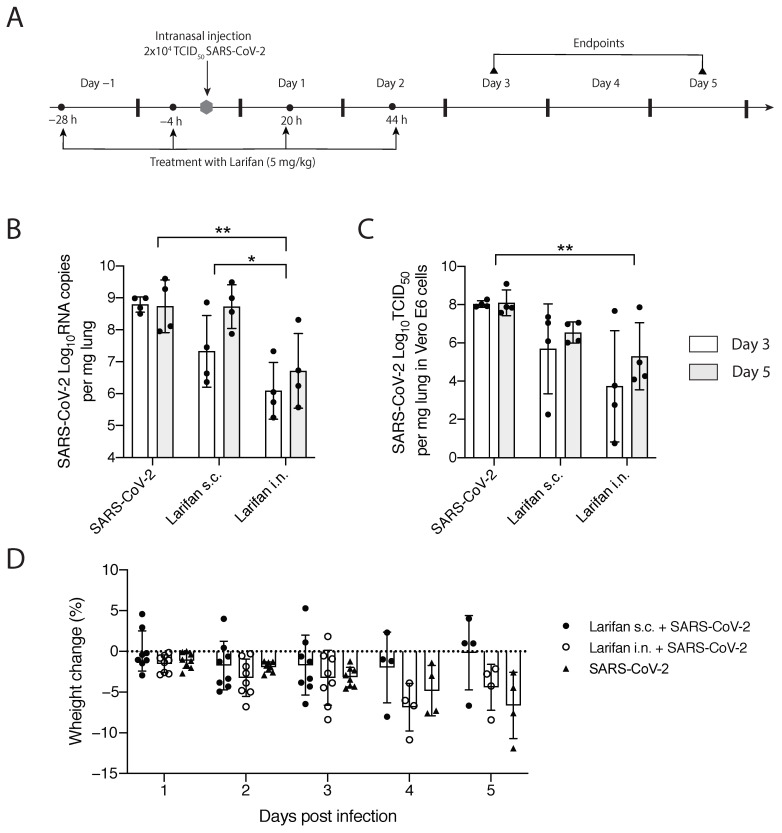
In vivo testing of Larifan’s effect in the SARS-CoV-2 infection model of golden Syrian hamsters. (**A**) Experimental strategy: hamsters were pre-treated with Larifan twice before virus infection, infected with SARS-CoV-2, and after infection treated two times more with Larifan always with an interval of 24 h between each administration. (**B**) Viral RNA copy number in the lungs of SARS-CoV-2 infected untreated and infected treated hamsters on days three and five after s.c. or i.n. Larifan administration. (**C**) Infectious virus titer in Vero E6 cells in the lungs of SARS-CoV-2-infected untreated and infected treated hamsters on days three and five after s.c. or i.n. Larifan administration. (**D**) Weight changes of SARS-CoV-2-infected untreated and infected Larifan-treated hamsters during the study. Data presented as mean (±SD). Statistical significance analysis was performed by using two-way ANOVA. * *p* < 0.05, ** *p* < 0.01. Dots represent individual hamsters.

**Figure 4 pharmaceuticals-15-01053-f004:**
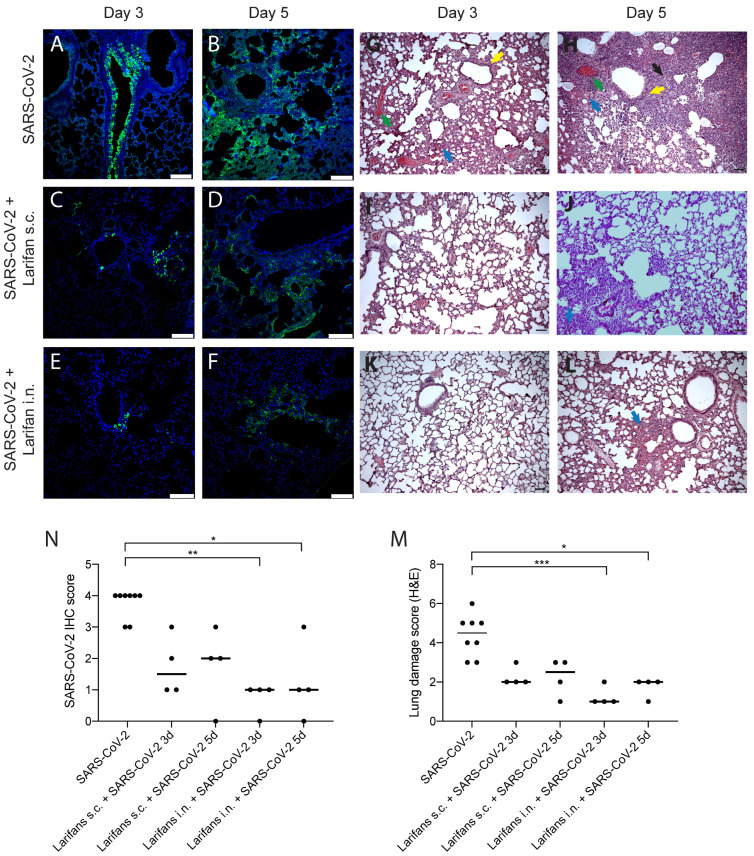
Larifan treatment reduces SARS-CoV-2 antigens in the lungs and improves lung histopathological features of infected animals. Representative immunohistopathology for SARS-CoV-2 NP (green) (**A**–**F**) and hematoxylin- and eosin-stained (**G**–**L**) images of lungs of infected and infected Larifan-treated hamsters. (**A**) Lungs of infected untreated hamster three days post-infection; immunoreactivity predominantly located in the bronchiolar epithelium. (**B**) Lungs of infected untreated hamster five days post-infection; immunoreactivity observed in different alveolar regions with patchy distribution pattern. (**C**,**D**) Lungs of infected and treated hamster three and five days after Larifan s.c. administration. (**E**,**F**) Lungs of infected and treated hamster three and five days after Larifan i.n. administration. (**G**,**H**) Lungs of infected untreated hamster three and five days post-infection: thickening of the interalveolar septa due to inflammatory infiltrate (blue arrow), vascular thrombosis (red arrow), alveolar septum fibrosis (black arrow), and signs of the bronchial epithelium hyperplasia (yellow arrow). (**I**,**J**) Lungs of infected and treated hamster three and five days after Larifan s.c. administration. (**K**,**L**) Lungs of infected and treated hamster three and five after Larifan i.n. administration. (**M**) The score of immunolabeled virus NP in lungs of infected untreated and infected Larifan-treated hamsters. (**N**) Histopathological damage severity score of hamster lungs of infected untreated and infected Larifan-treated animals. Data presented as mean. Statistical significance analysis was performed using Kruskal–Wallis tests with Dunn’s multiple comparison test. * *p* < 0.05, ** *p* < 0.01, *** *p* < 0.001. Dots represent individual hamsters. Scale bars 100 µm.

## Data Availability

Data is contained within the article.

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
