# Peer review of "Bacteriophage-Derived Double-Stranded RNA Exerts Anti-SARS-CoV-2 Activity In Vitro and in Golden Syrian Hamsters In Vivo"

_pharmaceuticals, 2022, doi:10.3390/ph15091053_

Round 1

Reviewer 1 Report

Authors aimed to study the antiviral activity of Larifan or Bacteriophage-derived dsRNA, against SARS-CoV-2. Authors found inhibition of SARS-CoV-2 replication in vitro and the reduction of the lung viral load/improved lung histopathology in infected hamsters treated with Larifan. The data suggests a potential use of Larifan in controlling the COVID-19 disease in humans. Overall, the manuscript is clearly written, the methods well designed, and the experimental data generally supports the main conclusions of the paper. However, I have some concerns that should be addressed prior to any publication.

  1. Figure 2, all concentrations of Larifan tested show a similar effect in vitro. Could the authors hypothesize why they did not see a dose response effect?
  2. Could the authors hypothesize why Full-time treatment does not show an effect in vitro, if both pre- and post- treatment show an effect?
  3. Could the authors hypothesize what would happen to past day 5 post infection in the hamster model? Do the treated hamsters clear the disease or does the drug just a short-term effect, as viral copies and titers are increasing again at day 5? Why were the end points chosen to be Days 3 and 5?
  4. Is the dosage used in the hamster experiments similar to the dose that is used in human studies?

Reviewer 2 Report

I would like to thank for the opportunity to review this paper.

This paper reported an antiviral drug of dsRNA, Larifan, as a potential anti-COVID-19 medication. The authors found that this drug significantly inhibits the SARS-CoV-2 replication both in vitro and in vivo. They observed that viral titer and RNA copies were reduced after the treatment of this drug in vitro in the human lung adenocarcinoma (Calu3) cells, and in vivo in Golden Syrian hamsters. However, as the authors noted (line 219-224), the underlying mechanism of its antiviral effects is still elusive. As the authors stated, Larifan has been obtained biotechnologically from E. coli cells infected with f2sus11 amber mutant bacteriophage and comprises dsRNA molecules (acidum ribonucleinicum duplicatum) with an average length of 700 base pairs, however, no description of this part in the methods and the sequence of the dsRNA Larifan is also not provided in this paper.

Some other issues also need to be further addressed.

 Figure 1C. Viral RNA copy number measurements performed in duplicate in Calu3 cell supernatants. Day 3, the SARS-CoV-2 RNA copies per µl is more than 6*106/µl. is this unit correct?

 Figure 2. why the full-time treatment is less effective than the other two conditions?

 Figure 3A, why not separate the conditions of the pre-infection and post-infection into two groups? And what’s the reason for the time points (-28, -4, 20, and 44 h) of Larifan treatment? And, from the results shown in figure 3B and 3C, Day 5 has a higher RNA copies and viral titer, suggesting that Larifan works for a short time.

Materials and Methods

 Infection of cells with SARS-CoV-2 and treatment with Larifan: 1. Cells were treated with varying concentrations (50, 100, 200 and 400 μg/ml) of Larifan. Is the concentration the final concentration? 2. What is the concentration of drugs in drug-containing medium? 3. For post-infection treatment, Larifan was added immediately after virus infection and maintained in the medium until the end of the experiment. For the post-infection treatment, why not add Larifan sometime (such as several hrs) after the infection of the virus? 4. For the pre-infection and post-infection treatment, will fresh medium replace the drug containing medium after a certain period of treatment (how long time for the treatment of the drug)?

 SARS-CoV-2 infection model in hamsters: could you provide more details about the groups of the 24 Golden Syrian hamsters? (SARS-CoV-2, Larifan s.c., Larifan i.n., 3days, 5days…)

 RNA isolation and viral RNA quantification by digital-droplet PCR (ddPCR) analysis: Viral copy numbers were assessed via digital-droplet PCR (ddPCR) using SARS-CoV-2 ddPCR kit (#12013743) from Bio-Rad with N1 and N2 primers and probes targeting the nucleocapsid genes. Please provide the sequences of N1 and N2 primers. Whether the results will be influenced by the mRNA of nucleocapsid gene?

 The score of immunolabeled virus NP and histopathological damage severity in lungs maybe too subjective.

Round 2

Reviewer 2 Report

I think the authors addressed my questions well and the paper is suitable for publication.